# Rapid heating rates define the volatile emission and regolith composition of (3200) Phaethon

**Martin D. Suttle** [1,2] ✉, **Lorenz. F. Olbrich** [3], **Charlotte. L. Bays** [2,4] **& Liza Riches**[1]

Asteroid (3200) Phaethon experiences extreme solar radiant heating (~750 °C) during perihelion (0.14 au), leading to comet-like activity. The regolith composition and mechanism of volatile emission are unknown but key to understanding JAXA's DESTINY⁺ mission data (fly-by in 2029) and the fate of near-Sun asteroids more generally. By subjecting CM chondrite fragments to fast, open system, cyclic heating (2-20 °C/min), simulating conditions on Phaethon we demonstrate that rapid heating rates combine with the low permeability, resulting in reactions between volatile gases and decomposing minerals. The retention of S-bearing gas limits the thermal decomposition of Fe-sulphides, allowing these minerals to survive repeated heating cycles. Slow escape of S-bearing gases provides a mechanism for repeated gas release from a thermally processed surface and, therefore the comet-like activity without requiring surface renewal to expose fresh material each perihelion cycle. We predict Phaethon regolith is composed of olivine, Fe-sulphides, Ca-sulphates and hematite.

Radiant heating from solar radiation significantly affects the exposed surfaces of airless objects, driving volatile gas emissions on small bodies (e.g.,[1]). Perihelion heating of comets adds significant quantities of dust into the inner solar system, while the more subtle effects on asteroid surfaces are often overlooked, despite the occurrence of 95 near-Sun asteroids passing within 0.2 au of the Sun during their perihelion approach[2] and the prevalence of short-duration thermal metamorphism in unshocked carbonaceous chondrites[3].

The near-Sun asteroid population includes the 5.6 km diameter object (3200) Phaethon (hereafter denoted without the prefixing numerical identity), the target of JAXA's DESTINY⁺ mission[4], the parent body of the Geminids meteor shower[5] and the focus of this study. Dynamical simulations indicate that Phaethon originated in the main asteroid belt, as a member of the Pallas asteroid family[6–8]. Its low geometric albedo (~6% [9] and blue-sloped, concave and featureless visible-near-IR (VNIR) spectrum are consistent with a B-type spectral class, related to the carbonaceous chondrites, and mostly closely matching the CI and CM chondrites[10–12]. However, the absence of a

3-micron feature[13] suggests an anhydrous composition, (and therefore no hydrated phyllosilicate minerals exposed at the asteroid's surface). More recently, work by MacLennan and Granvik[14] compared the mid-IR spectrum of Phaethon to carbonaceous chondrites and argued for a link to the CY chondrites. These are a newly described group of carbonaceous chondrites, characterised by aqueous alteration overprinted by short-lived thermal metamorphism resulting in dehydrated mineralogy (former phyllosilicates recrystallised to olivine). A unique feature of this group is the presence of Fe-sulphides at high abundances (> 20 vol%) (King et al. 2019[15];).

At perihelion, Phaethon's surface reaches temperatures up to 730 °C[6], sufficient for the thermal decomposition and volatile emission from chondritic mineral phases (e.g.,[16]). Correspondingly, Phaethon displays comet-like activity with brightening near perihelion, associated with gas emission[17,18] and with the presence of a faint tail extending ~250,000 km from the nucleus[19]. However, the decomposition mechanism and mineral phase responsible for Phaethon's emission activity remains ambiguous. Several potential explanations

[1]School of Physical Sciences, The Open University, Walton Hall, Milton Keynes, UK. [2]Planetary Materials Group, Natural History Museum, Cromwell Road, London, UK. [3]Department of Materials, University of Oxford, Parks Road, Oxford, UK. [4]Department of Earth Sciences, Royal Holloway University of London, Surrey, UK. ✉e-mail: martin.suttle@open.ac.uk

have been proposed, including the dehydration of hydrated silicates buried within the subsurface[20], the thermal decomposition of Fe-sulphides and carbonates[14], the release of refractory organic matter[21] and the decomposition of Na-bearing minerals (e.g., sodalite and nepheline) resulting in the emission of Na ions[16,18,22]. These reactions may be enhanced by physical processes such as thermal fracturing[6,23] and rotation-induced dust shedding/ejection events[24].

Understanding the fundamental geological processes operating within the thermal skin of Phaethon is crucial to interpreting data from DESTINY[+] and improving our knowledge of asteroid composition, surface dynamics, and the thermal evolution of near-sun objects. Laboratory experiments provide the opportunity to simulate solar radiant heating and test models of their geological processing. Previous heating studies on hydrated carbonaceous chondrites revealed one-way irreversible changes in composition arising due to the thermal decomposition of volatile-bearing minerals (e.g.,[25]). However, remarkably few studies have performed experimental heating under repeated heating-cooling cycles, which reflect the diurnal temperature fluctuations of near-sun asteroids. Patzek et al.[23] explored the physical effects of thermal cycling and demonstrated that thermal fatigue fractures develop rapidly (within 20 cycles) and are controlled by the abundance of hydrated minerals. Building on previous studies, we set out to investigate the activity of Phaethon through an experimental heating study. We performed repeated heating-cooling cycles using a Phaethon analogue material (Table 1), measured their mass change evolved gas release (Supplementary Table. S1) and studied their mineralogy and texture under a scanning electron microscope (SEM). Our data provide an empirical framework for understanding the activity of Phaethon.

Whole rock chips (36–60 mg and ~3-4 mm diameter) of the Murchison meteorite (a CM chondrite) were used as a Phaethon analogue. CM chondrites were deemed the most suitable analogue material for Phaethon, based on the dynamical and spectral link between Phaethon and the Pallas family. The Murchison CM chondrite is composed of hydrated silicates (Fe- and Mg-bearing phyllosilicate [~73 vol%]), anhydrous silicates (olivine and pyroxene [~23 vol%]), Fe-sulphides (tochilinite and troilite [~2 vol%]), carbonates (calcite [~1 vol%]) and Fe-oxides (magnetite [<1 vol%])[26].

Phaethon has a rotational period of 3.604 h[27], peak temperatures during perihelion reach up to ~730 °C (~1000 K) and heating/cooling rates are on the order of 10–25 °C/min[6]. Our selection of peak temperatures 500–750 °C (773–1023 K) and heating/cooling rates between 2–20 °C/min was designed to directly mirror the range of heating scenarios applicable to surface and shallow subsurface environments on Phaethon. We included single heating and repeated heating events, allowing the effects of cyclic thermal regimes to be discerned. In total, we conducted seven heating experiments (Table 1).

## Results

### Mass loss and volatile gas emissions

The Murchison CM chondrite, when heated to 750 °C lost between 11.4–16.7 wt.%. Correcting to remove the contribution from adsorbed terrestrial water[28] requires consideration of a thermal range 200-750 °C. This corresponds to a budget between 8.7-11.7 wt.% (Fig. 1, Table 2 and Supplementary Figs. S1–3). Variability in the amount of emitted volatiles is due to the heterogenous nature of CM chondrites, which are breccias, composed of multiple clasts and lithologies with variable degrees of aqueous alteration[29]. This mass loss range is consistent with previous studies investigating the volatile budget of CM chondrites (e.g.,[3,25]).

Double heating (two heating events with the same $T_{max}$) showed first cycle dominated volatile release (> 96 %), while the second cycle had minimal emissions (< 4 %). In the stepped heating experiments (two heating events where $T_{max}$ was 500 °C in the first heating cycle and then increased to 750 °C in the second heating cycle), roughly equal quantities of volatile gases were released in both heating cycles.

Only four gas species were identified (Table 2): $H_2O$ (3.0–10.3 wt.%), $CO_2$ (0.9–4.1 wt.%), $SO_2$ (0.1–1.4 wt.%) and organic matter, here represented by the most abundant molecule propene [$C_3H_5$] (0.2–0.6 wt.%); a reaction product of methylidyne radicals (CH) and ethane ($C_2H_6$)[30] involving the liberation of hydrogen. The latter is a crucial process for the formation of benzene, the simplest aromatic macromolecule and the building block of PAHs (polyaromatic hydrocarbons) in the interstellar medium. Water dominates the volatile emission (39–90 %), while $CO_2$ is subordinate (7–45 %), and $SO_2$ and organic matter are accessory phases (1–14 % and 1-2 % respectively). Each gas has a distinct emission profile, with $H_2O$ having three events, $CO_2$ being emitted above 200 °C, $SO_2$ being emitted at > 300 °C, and $C_3H_5$ being emitted across the entire heating range. A more detailed full description of the TGA-MS results is given in the supplementary materials.

### External surfaces

The unheated control chip and the single-heated chip (Sample E) appear black. In contrast, chips with >1 heating cycle (Samples A–D, F, G) either appeared red or mottled black and red. Their colour in optical light correlates with the Raman spectra collected on their external surface. Spectra from the black areas (Supplementary Fig. S4) contain first-order D (defect) and G (graphite) bands at ~1350 cm$^{-1}$ and ~1580 cm$^{-1}$, respectively, indicating the presence of macromolecular organic matter (e.g.,[31]). Conversely, the red regions (Supplementary Fig. S4) produced high fluorescence. These spectra lacked D and G bands (indicating an absence of organic matter) and instead exhibited peaks corresponding to the presence of haematite.

**Table 1 | Outline of experimental parameters investigated in this study**

| Sample ID | Mass (mg) | No. of cycles (N) | Ramp (°C/min) | 1st cycle: peak temperature (°C) | 2nd cycle: peak temperature (°C) | Description |
|---|---|---|---|---|---|---|
| Reference | 25 | 0 | N/A | N/A | N/A | Unheated reference |
| E | 48 | 1 | 2 | 750 | N/A | Single heating, slow rate |
| F | 52 | 2 | 10 | 750 | 750 | Double heating, intermediate rate |
| D | 45 | 2 | 10 | 750 | 750 | Double heating, intermediate rate (vacuum) |
| G | 59 | 2 | 20 | 750 | 750 | Double heating, fast rate |
| A | 36 | 2 | 10 | 500 | 750 | Double-stepped profile, intermediate rate |
| C | 39 | 2 | 20 | 500 | 750 | Double-stepped profile, fast rate |
| B | 39 | 8 | 20 | 750 | 750 | Cyclic heating, intermediate rate (physical test) |

Double heating refers to a scenario in which the $T_{max}$ was the same for both heating cycles, while a double stepped heating refers to a scenario in which the $T_{max}$ of the first cycle was lower than the $T_{max}$ of the second cycle.

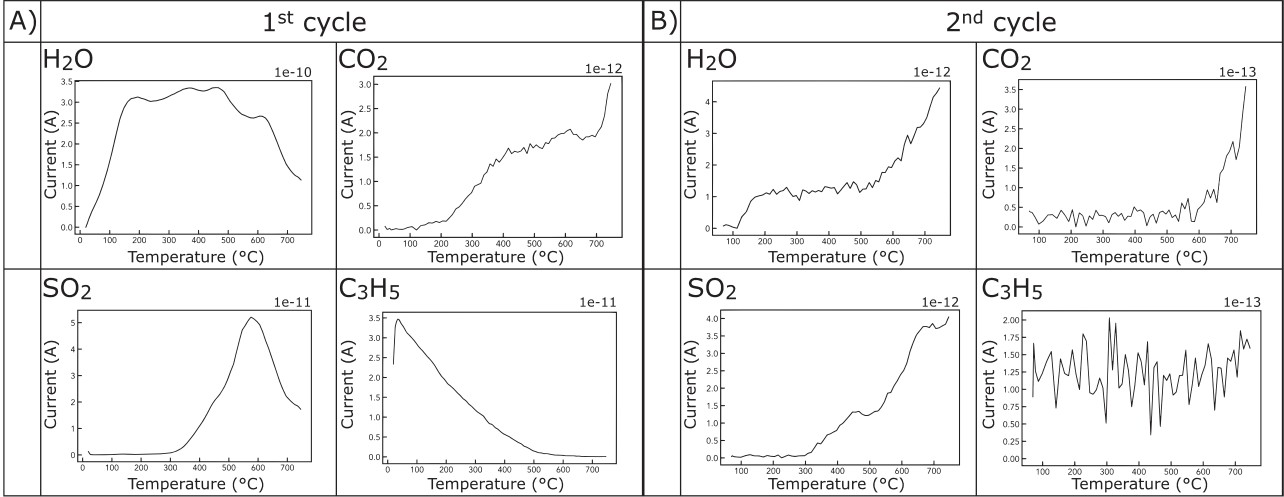

**Fig. 1 | Example TGA-MS, showing data for sample F (2x cyclic heating at the fast rate 20 °C/min).** Data are plotted as temperature vs. current (a proxy for the amount of emitted gas) profiles. Data are split into two columns (**A** & **B**) reflecting the 1st (**A**) and 2nd (**B**) heating cycles, respectively. The first row shows emission from the two major gas species ($H_2O$ and $CO_2$), and the second row shows emission from the two minor gas species ($SO_2$ and $C_3H_5$). Note: numbers shown in the top right-hand corner of each plot correspond to the correction factor applied to the Amperes y-axis.

## Microanalysis of the heated chips – sectioned surfaces

The heated chips (exposed surface areas of ~10 mm²) closely resembled the unheated reference but with increased fractures (with length scales of 100 μm to >1000 μm) (Supplementary Fig. S5). The presence of fractures is consistent with past heating experiments on hydrated carbonaceous chondrites and occurs due to a combination of phyllosilicate dehydration and thermal fatigue[23,32,33].

Elemental mapping revealed no systematic change in elemental distribution (relative to the unheated starting material), except for sulphur, oxygen and sodium. In unheated CM chondrites, sulphur occurs primarily within Fe-sulphides where it occurs as pyrrhotite-troilite and as tochilinite cronstedtite intergrowths (TCIs), with lesser quantities of pentlandite and submicron-scale Fe-Ni-sulphide grains dispersed within the fine-grained matrix[34,35]. Changes in the distribution of sulphur in the heated chips (Fig. 2) primarily affected the TCIs and the smallest Fe-sulphides, dispersed within the fine-grained matrix. In the unheated reference, these sulphides were heterogeneously distributed, and sulphur abundances within the unheated fine-grained matrix ranged from 1.0-6.2 wt.%, averaging 2.7 ± 1.28 wt.% (1 σ, N = 55). By contrast, in the single-heated experiment (sample E), sulphur abundance in the fine-grained matrix was more homogenously distributed (S: 2.5 ± 0.45 wt.% [1 σ], N = 15) and a thin (30–60 μm) sulphur depleted zone along the chip's perimeter was observed (Fig. 2B). Similar but more pronounced S-depletions are observed in the cyclically heated samples. For example, the heating of Sample F (double heating at the intermediate rate) resulted in a homogenous S distribution in the particle centre (S: 2.5 ± 0.96 wt.% [1 σ], N = 22) and a larger (~130–250 μm) S depleted rim region (S: 0.3 ± 0.24 wt.% (1 σ, N = 37)). Heating in samples A, C, D and G produced even thicker S depleted rims (typically >300 μm). Furthermore, in several of these samples the S-poor regions also extended inwards to the chip's centre along the margins of the thermal fractures (Fig. 2C). Sample B, which was subjected to eight heating cycles, showed low overall sulphur abundance within the fine-grained matrix (S: 1.7 ± 0.49 wt.% [1 σ], N = 21), but sulphur enrichment within the rim region (Fig. 2D), making its sulphur distribution distinct from that of all the other heated samples.

Elemental mapping of the oxygen distribution revealed a slight increase near the chip's edge in all twice-heated samples, this is interpreted as the formation of haematite dispersed within the fine-grained matrix. Elevated oxygen abundance is anti-correlated with sulphur abundance (Supplementary Fig. S6).

Elemental mapping of the sodium distribution reveals subtle changes in concentration across the samples. In the unheated reference chip, Na is homogenously distributed within the fine-grained matrix. EDS analyses demonstrate low abundances, Na: 0.3 ± 0.11 wt.% (1 σ, N = 57). In the heated samples, Na likewise occurs at low abundances (typically < 0.5 wt.%), although localised enrichments occur, typically adjacent to chondrules, along the margins of radial fractures within chondrule fine-grained rims or at the edge of the sample (Supplementary Fig. S7).

Carbonates are an accessory phase in CM chondrites (< 0.5–4.0 vol%), where they occur primarily as calcite[26]. In the unheated reference sample, carbonates are relatively rare, reflecting the mild extent of aqueous alteration recorded by the Murchison meteorite (CM2.6[36],). They predominantly occur as small grains of calcite mantled by Fe-rich rims composed of tochilinite-cronstedtite intergrowths (termed TCIs). They are termed "T1" calcites and interpreted as an early generation of carbonates (e.g.,[37]) (Fig. 3A). In the heated samples, intact calcite is rare. Instead, former carbonates appear as anhedral grains or grain aggregates (Fig. 3B–D). Geochemical analyses (Supplementary Table S2 and Supplementary Fig. S8) reveal variable compositions, reflecting a mix of calcite and Ca-sulphate (anhydrite). Raman spectra on these phases revealed peaks corresponding to anhydrite.

## Discussion

Carbonaceous chondrite fine-grained matrix is characterised by low permeability, as demonstrated by Corrigan et al.[38] and modelled by Bland et al.[39], with estimates between $10^{-19}$ to $10^{-16}$ m² (0.1–100 μD). However, during heating, permeability increases due to dehydration cracks and thermal fractures, allowing the escape of some volatile gases, as detected by the TGA-MS experiments.

Analysis of the heated samples revealed zones of sulphur depletion along margins and adjacent to fractures but retention of Fe-sulphides within the centre of the heated chips (Fig. 2). Therefore, Fe-sulphides in rim regions and near fractures suffered thermal decomposition and loss of sulphur-bearing gas, while those located more centrally within the chip survived.

The nominal thermal decomposition temperatures of tochilinite and pyrrhotite-troilite minerals lie between 200–500 °C[40,41]. Thus, the entire Fe-sulphide budget may be expected to decompose in all of the experiments we ran ($T_{max} = 750$ °C). However, the thermal

**Table 2 | Quantified volatile emission data for samples A–G (quoted in wt.% to 3 decimal place precision)**

| Sample | Sample E | Sample F | Sample D | Sample G | Sample A | Sample C |
|---|---|---|---|---|---|---|
| Heating pattern | Single | Cyclic | Cyclic | Cyclic | Stepped | Stepped |
| Rate (°C/min) | Slow, 2 | Intermediate, 10 | Intermediate, 10 | Fast, 20 | Intermediate, 10 | Fast, 20 |
| Carrier gas | Ar | Ar | Vacuum | Ar | Ar | Ar |
| Step 1 (°C) | 20–200 | 30–200 | 30–200 | 20–200 | 25–200 | 25–200 |
| $H_2O$ (wt.%) | 3.839 | 2.844 | 4.504 | 2.424 | 2.422 | 2.578 |
| $CO_2$ (wt.%) | 0.235 | 0.021 | 0.571 | 0.066 | 0.024 | 0.098 |
| $SO_2$ (wt.%) | 0.008 | 0.005 | 0.085 | 0.002 | 0.004 | 0.021 |
| $C_3H_5$ (wt.%) | 0.503 | 0.101 | 0.070 | 0.073 | 0.189 | 0.224 |
| Subtotal (wt.%) | 4.585 | 2.970 | 5.230 | 2.565 | 2.639 | 2.920 |
| Step 2 (°C) | 200–750 | 200–750 | 200–750 | 200–750 | 200–500 | 200–500 |
| $H_2O$ (wt.%) | 8.903 | 6.536 | 9.687 | 4.023 | 1.135 | 2.234 |
| $CO_2$ (wt.%) | 2.460 | 3.488 | 0.792 | 3.758 | 1.835 | 1.475 |
| $SO_2$ (wt.%) | 0.118 | 1.258 | 0.133 | 0.649 | 0.501 | 0.365 |
| $C_3H_5$ (wt.%) | 0.072 | 0.115 | 0.143 | 0.107 | 0.140 | 0.043 |
| Subtotal (wt.%) | 11.552 | 11.398 | 10.755 | 8.536 | 3.610 | 4.117 |
| Step 3 (°C) | 750–40 | 750–70 | 750–70 | 750–140 | 500–70 | 500–140 |
| $H_2O$ (wt.%) | ND | ND | ND | 0.010 | ND | ND |
| $CO_2$ (wt.%) | ND | ND | ND | 0.005 | ND | ND |
| $SO_2$ (wt.%) | ND | ND | ND | 0.001 | ND | ND |
| $C_3H_5$ (wt.%) | ND | ND | ND | ND | ND | ND |
| Subtotal (wt.%) | ND | ND | ND | 0.015 | ND | ND |
| Step 4 (°C) | N/A. | 70–750 | 70–750 | 140–750 | 70–750 | 140–750 |
| $H_2O$ (wt.%) | | 0.108 | 0.418 | 0.082 | 2.379 | 1.798 |
| $CO_2$ (wt.%) | | 0.123 | 0.040 | 0.072 | 2.240 | 2.191 |
| $SO_2$ (wt.%) | | 0.116 | 0.002 | 0.092 | 0.718 | 0.520 |
| $C_3H_5$ (wt.%) | | 0.005 | 0.003 | 0.006 | 0.018 | 0.026 |
| Subtotal (wt.%) | | 0.352 | 0.464 | 0.253 | 5.355 | 4.535 |
| Step 5 (°C) | N/A. | 750–90 | 750–90 | 750–160 | 750–150 | 750–170 |
| $H_2O$ (wt.%) | | ND | 0.240 | 0.003 | ND | ND |
| $CO_2$ (wt.%) | | ND | 0.027 | 0.002 | ND | ND |
| $SO_2$ (wt.%) | | ND | 0.002 | 0.001 | ND | ND |
| $C_3H_5$ (wt.%) | | ND | 0.003 | ND | ND | ND |
| Subtotal (wt.%) | | ND | 0.271 | 0.005 | ND | ND |
| Grand total (wt.%) (steps 1–5) | 16.138 | 14.720 | 16.721 | 11.374 | 11.604 | 11.572 |
| Corrected total (wt.%) (steps 2–5) | 11.552 | 11.749 | 11.491 | 8.810 | 8.965 | 8.652 |

(Note TGA-MS data were not collected for the 8x heated sample B). Data are split into five steps: step 1 covers the initial low-temperature heating (~ 20–200 °C), step 2 covers the remainder of the initial heating ramp, step 3 covers cooling at the end of the 1st heating event, step 4 covers the second heating ramp and step 5 covers cooling at the end of the 2nd heating event. Total volatile emissions are calculated as the sum of steps 1–5. However, since carbonaceous chondrite meteorites contain a non-negligible component of adsorbed terrestrial water (e.g.,[28]), the corrected mass loss, after removal of terrestrial water, is calculated as the sum of steps 2–5. For all experiments, $H_2O$ and $CO_2$ dominate the volatile release, while $SO_2$ and organic matter represent minor components in the volatile inventory. "ND" refers to "not detected" for a given gas species over that temperature range.

decomposition behaviour of Fe-sulphides is complex, being dependent upon both temperature and $fS_2$ fugacity. As temperatures rise, pyrrhotite-troilite group minerals progressively decompose, evolving from sulphur-rich pyrrhotite (Fe$_{[1-x]}$S, where $x = 0.0–0.2$) towards stochiometric troilite (FeS) before fully decomposing to Fe-metal + $S_2$ gas:

$$2FeS \rightleftharpoons Fe + S_2 \tag{1}$$

Under closed system conditions, the release of sulphur gas from Fe-sulphides that decompose early (tochilinite and potentially some pyrrhotite) increases the $fS_2$ of the system. Elevated $fS_2$ is then able to stabilise the remaining sulphide minerals, inhibiting their thermal decomposition, even as temperatures rise far beyond their open system stability field[40,42,43]. In our experiments, the rim regions and matrix adjacent to large cracks allowed the escape of sulphur-bearing gases and, therefore, the entire Fe-sulphide complement in these locations

decomposed (this was detected by the TGA-MS, [Table 2 and Supplementary Figs. S1–3]). By contrast, in the centre of most chips, Fe-sulphide survival is attributed to the slow (relative to the length of the heating experiment) release of liberated sulphur-bearing gases (and their buffering effects).

Where sulphur-bearing gases remain within the system during retrograde cooling, these gases can back-react with the heated mineralogy to form a new generation of sulphur-bearing minerals. The CY chondrites record a short-lived post-hydration thermal metamorphic event[15], this process led to the retention of abundant stochiometric troilite despite peak metamorphic temperatures estimated up to 750 °C. In the CY chondrites, back-reacting sulphur-bearing gas produced a new generation of Fe-sulphides, some of which were located within the dehydration cracks of phyllosilicates, further demonstrating their formation after heating[40]. In our experiments, the same effect explains observations of a more homogenous sulphur

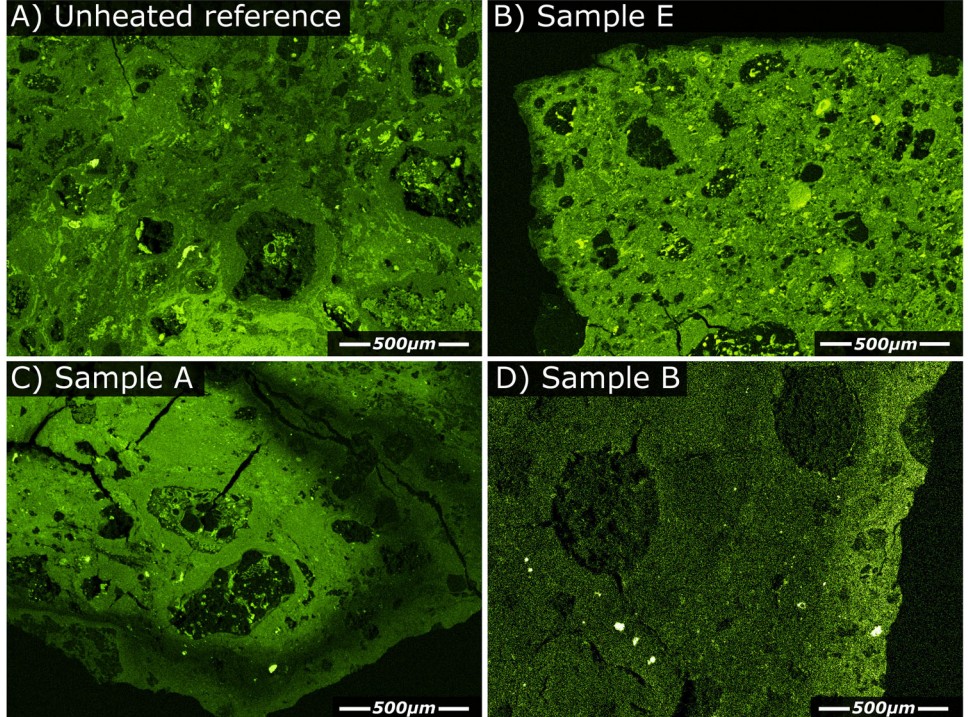

**Fig. 2 | Single element EDS maps showing the distribution of sulphur (S-K line).** Sulphur is heterogeneously distributed in the unheated reference (**A**). Heating initially leads to homogenisation of the S-distribution, as demonstrated by the single-heated sample E (**B**). This sample also reveals a thin band of S-depletion along the chip's perimeter. By contrast, cyclic heating (**C**) results in more progressive S-depletion, forming a thicker S-depleted rim and the occurrence of S-depleted regions adjacent to thermal fractures that penetrate into the chip's interior. In the 8x heated sample B (**D**), the S-distribution is inverted relative to the 1x and 2x heating experiments. Here, the rim is S-enriched. Note these maps display only the relative intensity of S-abundance. Absolute S-abundance is highest in the unheated chip and lowest in the 8x heated chip.

distribution within the fine-grained matrix of heated chips relative to the unheated reference (Fig. 2).

Back reactions between sulphur-bearing gases and solid minerals also explain the occurrence of Ca-sulphate minerals, which are not otherwise found in unheated carbonaceous chondrites, nor previous experimentally heated carbonaceous chondrites. The thermal decomposition of calcite occurs at temperatures between 600–850 °C and is independent of $f\mathrm{CO_2}$[44]. Calcite decomposition releases $CO_2$ gas and forms quicklime (CaO):

$$CaCO_3 \rightarrow CaO + CO_2 \qquad (2)$$

However, quicklime is metastable and will readily back-react with residual gases during retrograde cooling to form a more stable phase. Within the CM chondrite Sutter's Mill, Haberle and Garvie[45] reported the formation of portlandite (Ca[OH]₂) and oldhamite (CaS) occurring due to back-reaction between quicklime (from calcites) and released gases from phyllosilicates ($H_2O$) and sulphides ($S_2$). They argued that these unusual Ca-bearing minerals were formed by thermal metamorphism on the Sutter's Mill parent body. The apparent absence of these minerals in the numerous other CM chondrites affected by post-hydration heating[3] was suggested to be due to terrestrial weathering (leading to their destruction). Peak temperatures in our experiments (< 750 °C) were within the carbonate decomposition range (600–850 °C). We identified mixed-phase calcium-rich aggregates composed of both carbonate and sulphate (Fig. 3 and Supplementary Table S2). They represent carbonates that suffered partial thermal decomposition. The resulting quicklime back-reacted with S-bearing gases to form Ca-sulphates.

Mass spectrometer analysis revealed that sulphur-bearing molecules were exclusively sulphur monoxide fragment ions ($SO^+$ and $SO^{2+}$), implying $SO_2$ as the parent molecule. The absence of detectable $S_2$ gas (released from the thermal decomposition of Fe-sulphides) suggests oxidation of sulphur gas, likely by water released from decomposing phyllosilicates, demonstrating volatile gas interaction during heating. Under these conditions, the expected back-reaction would be:

$$CaO + SO_2 + \tfrac{1}{2}O_2 \rightleftharpoons CaSO_4 \qquad (3)$$

Small amounts of organo-sulphur compounds, associated with the diffuse organic matter held in a chondrite fine-grained matrix, could also contribute to some $SO_2$ gas[46]. Eq. 3 explains why our experiments produced Ca-sulphates, instead of the Ca-sulphide (oldhamite) observed by Haberle and Garvie[45]. The higher $fO_2$ of our system was most likely due to the rapid liberation of $H_2O$ from phyllosilicates and its slow release from the chips. This would lead to partial retention of $H_2O$ during the later stages of heating and, therefore, more oxidising conditions. Thus, our data suggests that the two crucial factors in generating Ca-sulphates within a heated chondritic lithology are (1) a rapid heating rate and (2) low permeability, which combine to prevent residual volatile gases ($SO_2$ and $H_2O$) from escaping. These gases are then present during higher-temperature carbonate decomposition and are available to back-react with the quicklime.

There is evidence for similar back reactions between decomposing carbonates and volatile gases in the spectral data from Phaethon. MacLennan and Granvik[14] inferred the presence of portlandite (Ca(OH)₂) (and brucite [Mg(OH)₂]), which is best explained as the product of carbonate decomposition and back-reaction with water

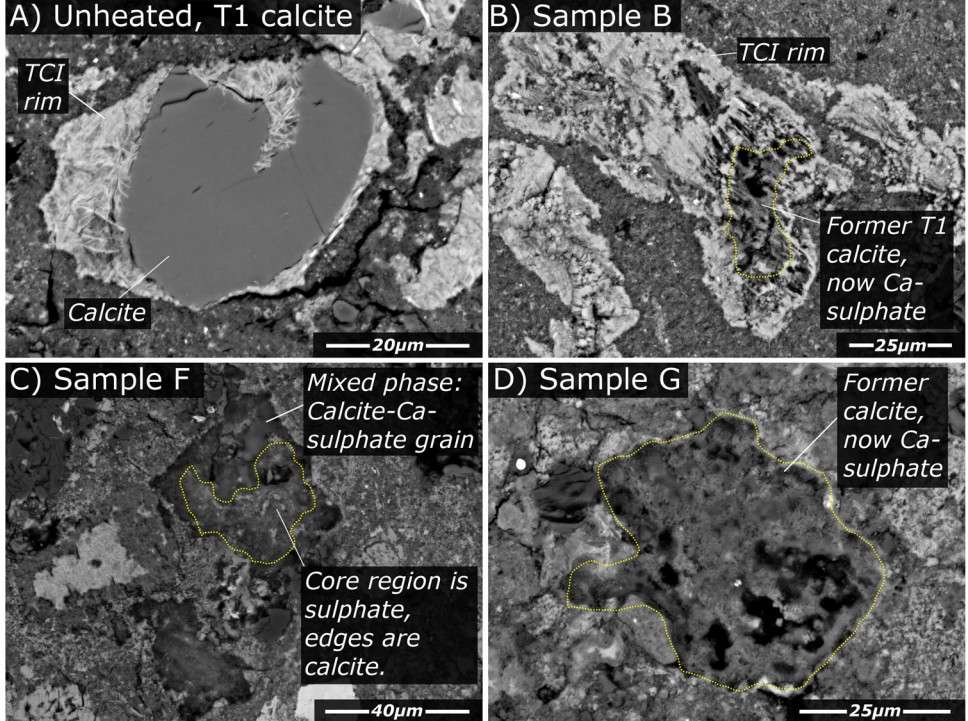

**Fig. 3 | The reaction between decomposing carbonates and S-bearing gases in the heated chips produced Ca-sulphate minerals. A** An example of a first-generation (T1 calcite) in an unheated CM chondrite. **B** An example of a former T1 calcite in Sample B (8x heated), now replaced by Ca-sulphate. **C** a mixed phase Calcite/Ca-sulphate grain in sample F (Double heating, intermediate rate), interestingly calcite is preserved along the margins of the grain while the core decomposed first and has since reacted to form Ca-sulphate. **D** A Ca-sulphate grain in Sample G (Double heating, fast rate). All the heated samples (A-E) contained Ca-sulphates. Note: the term TCI used in panels (**A**) and (**B**) denotes tochilinite-cronstedtite intergrowths.

released from phyllosilicates. The disparity between predictions of Ca-sulphate (this work) and Ca-hydroxide (spectral data) formation reflects either differences in the ratios of volatile gas species ($H_2O$ and $SO_2$) active at the time of back-reaction or could be because Ca-sulphate spectra were not included in MacLennan and Granvik's[14] spectral mixing model.

The formation of haematite, as identified in our experiments, also requires explanation. Elemental mapping revealed elevated oxygen abundance in a fine-grained matrix near chip edges. This was confirmed as haematite by Raman data. Elevated oxygen abundance is anti-correlated with sulphur abundance (Supplementary Fig. S6), demonstrating that the regions of Fe-sulphide retention (and Ca-sulphate formation) are distinct from the regions of haematite formation. These regions reflect closed-system heating and open-system heating, respectively. In open system regions where liberated volatile gases were able to escape quickly (near the chip edge and adjacent to large cracks), Fe-sulphides suffered complete thermal decomposition (Eq. 1), forming Fe-metal. This newly formed metal would be susceptible to oxidation during the retrograde cooling cycle. Escaping $H_2O$ and $CO_2$ liberated from phyllosilicates and carbonates (respectively) are capable of acting as oxidising agents, resulting in the conversion of Fe-metal to Fe-oxides. The fine-grained matrix in CM chondrites already contains magnetite at abundances between 0.5–5.0 vol%[26] and this is able to survive heating without decomposition (Supplementary Fig. S9). It is possible that some, or all of this magnetite within these regions was also oxidised to haematite.

Next, we consider whether sodium has a role in Phaethon's emission profile. Masiero et al.[16] proposed Na loss from feldspathoids (nepheline and sodalite) within a carbonaceous chondrite precursor as the possible source of Phaethon's activity. They performed numerical models of asteroid heating and combined this with experimental heating of the Allende CV chondrite, demonstrating up to 50% Na loss

from feldspathoids after heating to 800 °C. Likewise, Zhang et al.[18] identified fluoresce from Na D-lines (589.0/589.6 nm) in Phaethon's comet-like tail as imaged by the Large Angle Spectrometric Coronagraph (LASCO) instrument on the Solar and Heliospheric Observatory (SOHO). This suggests that Na ions are at least partially responsible for Phaethon's brightening and tail formation.

The experiments conducted in this study also found evidence for Na mobilisation and migration across the samples, with Na enrichment occurring along the margins of chondrules, close to fracture walls and at the edge of the sample on the twice-heated samples (Supplementary Fig. S7) However, Na ions were not detected in the MS data, indicating negligent Na emission (the detection limit for m/z values of 23 [Na] would be approximately > 0.02 wt.%). This is not surprising when the low abundance of Na in the starting material is considered.

In hydrated carbonaceous chondrites, Na is primarily located within the fine-grained matrix (the host mineral phase is not established but could be phyllosilicate), while in nominally dry chondrites, affected by metamorphism, Na is held primarily within feldspathoids (< 3.5 vol% in the CO chondrites[47] and <1 vol% in the CV chondrites [48].

The average bulk Na abundance measured in a range of carbonaceous chondrites is always low, < 0.4 wt.% (Braukmüller et al. 2018). On this basis, Na is unlikely to be the sole driver of Phaethon's activity and may be supplemented by additional gas species. By contrast, S occurs at abundances 2.0–3.5 wt.% in carbonaceous chondrites and at ~ 6.2 wt.% for the S-rich CY chondrites (based on analysis of Y-980115) (Braukmüller et al. 2018). This S is held in Fe-sulphide minerals and can provide an order of magnitude more mass to be released. Thus, we argue that S is probably a major volatile species driving activity on Phaethon, but there remains good evidence for the role of Na as well.

Having evaluated the response of the CM chondrite analogue materials to cyclic heating, we propose the following experimentally

constrained model for the regolith composition and activity of Phaethon. Near-identical magnitude brightening, reoccurring each orbit, has been observed around perihelion[49]. Recent high-resolution thermal IR imaging constrained an upper limit for the quantity of dust released from Phaethon (< 14 kgs⁻¹). This is estimated as 50x less than the rate required to sustain the Geminid meteor stream[50]. Furthermore, Zhang et al. [18] argued that the comet-like tail observed extending from Phaethon[19] is unlikely to be a dust tail but, instead, a (Na) ion tail. Thus, the role of surface renewal by dust shedding in the current era seems minimal yet Phaethon continues to demonstrate comet-like activity (gas emission). Continued activity from a surface that has been repeatedly heated over thousands of years (> 20,000 years [6] is what makes explaining the activity of Phaethon challenging. Most thermal decomposition reactions within a hydrated chondritic assemblage are irreversible (e.g.,[25]), volatile release occurs only once when temperatures first exceed the decomposition threshold. Therefore, any viable explanation for Phaethon's activity must explain how a thermally processed surface can continue to emit volatile gases each heating cycle.

MacLennan and Granvik[14] used a spectral mixing model to infer the surface composition of Phaethon: olivine (36 wt.%), Fe-sulphide (42 wt.%), carbonates (8 wt.%) and decomposition products of carbonates (hydroxide minerals) (16 wt.%). From this mineralogy, they suggested that the thermal decomposition of phyllosilicate (forming olivine), Fe-sulphide and carbonate were most likely the source of volatile gases ($H_2O$, $CO_2$ and $S_2$) driving Phaethon's activity. However, their model did not address how repeated emissions from a heated surface could occur.

In this study, we empirically investigated the cyclic heating of a Phaethon analogue (CM chondrite) under heating regimes similar to those experienced by Phaethon at perihelion and demonstrated the crucial role of reversible Fe-sulphide decomposition. When combined with low permeabilities that limit open system gas loss, this provides a viable mechanism for repeated emission with each heating cycle. As such, this work extends the model of MacLennan and Granvik[14], reducing the role of $H_2O$ from phyllosilicate decomposition and $CO_2$ from carbonate decomposition as insignificant beyond the first heating event and elevating S-bearing gases as the main volatile species in later heating events. Based on the data in this work, we suggest a generalised model for the emission of volatile gases, and the formation of regolith on primitive asteroids ejected into near-Sun orbits and applicable to Phaethon:

> During the first heating event, the unheated chondritic matrix experiences rapid thermal decomposition of hydrated secondary minerals (phyllosilicates, Fe-sulphides and carbonates) as well as significant thermal fracture. Low permeabilities inhibit the escape of volatile gases in zones distal to fractures. $H_2O$ released from phyllosilicate decomposition oxidises $S_2$ released from Fe-sulphide decomposition, forming $SO_2$ gas. As temperatures increase to 750 °C carbonates decompose, forming reactive quicklime and releasing $CO_2$. During the cooling path, residual $SO_2$ gas reacts with the CaO to form a new generation of Ca-sulphates. In addition, in regions where open system conditions prevail (near the surface and along fracture walls), haematite formation occurs. This is explained by the loss of S-bearing gases allowing the complete decomposition of Fe-sulphides and leaving a population of reactive Fe-metal (and pre-existing magnetite) that is oxidised by $H_2O$ and $CO_2$ to generate haematite.

> During subsequent heating cycles, the majority of the matrix mineralogy remains unchanged, being composed of stable, anhydrous olivine (with minor quantities of Fe-oxides, hydroxides and anhydrite). However, the remaining Fe-sulphide population will continue to decompose, releasing S-gas, and some of these gases will slowly escape the sample, while the remaining sulphur-bearing gases cool and back-react to form new sulphide and sulphate minerals. Ultimately, the Fe-sulphides will fully decompose to form

Fe-metal, at which point comet-like activity will cease until a fresh unheated chondritic surface is exposed by dust shedding, tidal fracture or another resurfacing mechanism.

Finally, we must consider whether Phaethon is best interpreted as either a CY or CM chondrite. MacLennan and Granvik[14] argued that Phaethon is likely composed of CY chondrite material: "…*the interior of Phaethon consists of relatively unaltered, hydrated CY 'precursor' material that has not been heated*". However, we suggest that the high Fe-sulphide abundance detected on the surface of Phaethon by mid-IR spectroscopy instead occurs primarily due to passive enrichment of Fe-sulphide (survival through multiple heating events), while the remaining hydrated minerals decomposed to form volatile-depleted anhydrous phases – olivine and lesser amounts of Ca-carbonate decomposition products (portlandite and/or anhydrite). Conversely, the high sulphide abundances in the CY chondrites originated either as an accretionary property (condensation from the solar nebula) or through aqueous alteration by S-rich fluids that upwelled from the interior of a partially dehydrated/differentiated parent body[15]. In addition, although the heat that drove phyllosilicate and carbonate decomposition in the CYs was short-lived (hours to days), the most likely mechanism is impacts and not solar radiant heating, as evidenced by heterogenous heating effects and the widespread presence of brecciation (abundant mm-scale clasts) (King et al. 2019[15,40];). Therefore, we conclude that the proposed similarities in mineralogy between Phaethon and the CYs do not necessarily argue for a direct match between these groups. Instead, dynamical evidence for the origin of Phaethon as a member of the Pallas family and the near-IR spectral link between Phaethon and CM/CIs[6–8] suggests that Phaethon is composed of CM or CI material but that its surface has been altered significantly by solar radiant heating.

## Methods
### Heating experiments
Heating experiments were performed using a thermogravimetric analyser (TGA) (STA 449 F3 Jupiter® coupled to a mass spectrometer (QMS 403 Aëolos Quadro)) based at the Henry Royce Institute, University of Oxford branch (Supplementary Fig. S10). All heating experiments used whole rock chips. Our experimental setup used alumina crucibles heated within an inert gas under laminar flow (argon at 20 mL/min). Sample mass, heat flow, and temperature were measured continuously. The released gas species were detected by a quadrupole mass spectrometer using bar graph measurement mode from m/z 1 to 65. A test run up to m/z = 300 was conducted to ensure no larger mass to charge fragments were released. Mass loss data were obtained from the TGA (given in wt.% with a precision of ± 0.0001 mg), while evolved gas species were identified using mass spectrometer m/z-values and look-up tables (Supplementary Table. S1). The abundance of each evolved gas species ($H_2O$, $CO_2$, $SO_2$ and organic matter [$C_3H_5$]) was determined as follows. The experimental runs were divided into temperature ranges (e.g., 0–200 °C, 201–400 °C, etc.). Over each range, we multiplied the integral of current*time by the m/z-value of the parent molecule. This corrected for the differing mass of each evolved gas species. Related integrals (e.g., 22 [$CO^{2+}$] plus 44 [$CO_2^+$]) were then summed to determine the contribution from each gas species. Next, the ratio of weighted integrals was divided by the mass loss fraction at each temperature range providing a weight percentage (wt.%) estimate for each gas species as a function of temperature (given as discrete thermal "windows"). This approach has the following underlying assumptions:

- There is a linear correlation between the m/z ion current and the amount of gas released with the slope of the linear correlation is similar amongst different fragments.
- The fragment analysis was simplified such that only m/z values with sufficient signal (0.0005 nA) were considered. This limited

our assessment to the major fragment ions and their parent molecules. As a result, abundance estimates of each gas species are minimum estimates.

- All mass loss was attributed only to the four main gas species ($H_2O$, $CO_2$, $SO_2$ and organic matter, predominantly propene [$C_3H_5$]).
- A simple baseline correction was applied to the mass spectrometer (current-time) data. This correction was required to compensate for background gas flow reaching the detector (the baseline correction subtracted the minimum current at each given section, which ensured that the ion current quantification considers only gas released by the samples and not the carrier gas [Ar]).

In addition to the heating experiments outlined above we performed a single experiment in which the rock chip (sample D) was heated under vacuum conditions, as opposed to an inert gas. The motivation for this experiment was to confirm that the inert gas did not affect the thermal decomposition behaviour and therefore, that our experimental design accurately simulated asteroid perihelion-induced heating scenarios. The inert gas is required as a carrier, ensuring volatiles released from the sample are effectively transported to the mass spectrometer and leading to temperature-resolved emission events that are accurately linked to gas detection events. Thus, under vacuum conditions, the mass spectrometer data is expected to be both noisier and affected by time-averaging.

### Microanalysis

Following experimental heating, the chips were analysed using Raman spectroscopy. We used a Jobin-Yvon Horiba LabRam HR (800 mm) Raman microscope, located at The Open University in Milton Keynes. This instrument utilises a 514.53 nm (green) laser as the excitation source, with the beam focused through a 100x objective lens. The slit width and confocal pinhole aperture were set to 200 and 100 μm, respectively, with 1800 grooves/mm grating used to disperse the Raman signal. The measured beam diameter and laser power at the sample surface were ~ 2 μm and ~ 12.6 s μW, respectively. For each point of interest, we used a 15 s exposure time (per spectrum) and three accumulations to lower the signal-to-noise ratio and remove any erroneous spectra, leading to a total acquisition time of 45 s.

Whole-section EDS maps were collected on a Zeiss EVO LS15 scanning electron microscope (SEM), fitted with an Oxford Instruments' 80 mm² X-Max silicon drift detector energy dispersive spectrometer (EDS), and located at the Natural History Museum (NHM), London. Maps were generated by montaging multiple individual fields (~ 60–100 fields per sample), each composed of approximately 550 μm x 415 μm, with a pixel size of ~ 1.5 micrometres. Maps were collected at 20 kV, with a 1.5 nA beam current. Each field had a total acquisition time of ~ 270 s. Maps were used to explore the intra-sample variability, coarse-scale texture, and the distribution of mineral phases.

Quantitative chemical data (EDS spot analyses) were also obtained using a different SEM (a TESCAN Clara SEM) located at The Open University. This instrument is fitted with an Oxford Instruments Ultimax 170 detector and provides standardless EDS data. Prior to analysis, the EDS system was calibrated (peak position and intensity) using a pure copper metal reference. All spot analyses were performed at high-vacuum, with an accelerating voltage of 20 kV, a beam current of 1.5 nA and a fixed optimal working distance (10 mm). We used the Oxford Instruments Aztec software (version 6.1 HF3), and the following EDS acquisition settings were applied: process time 4, 2048 energy channels, and a live acquisition time of 20 s. Output count rates for silicate minerals ranged between 70–90 kcps, which on this detector corresponded with a 50 % deadtime. Data were processed with standard XPP matrix correction routines. For silicates, oxides, sulphates and analyses within the chondritic fine-grained matrix, we applied the "oxygen by stoichiometry" quantification routine. For analyses of metal and sulphides, we applied the "all elements" quantification routine. All geochemical data are given as uncorrected weight totals (wt.%) and quoted to one decimal place.

## Data availability
All data generated in this study are provided in the manuscript itself and within the supplementary information file.

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

## Acknowledgements

C.L.B. is supported by the Natural Environment Research Council and the ARIES Doctoral Training Partnership (grant number NE/S007334/1). The NHM, London, provided a destructive loan of Murchison meteorite material (loan number: MIN2022-553 MET [16/09/22]).

## Author contributions

M.D.S. conceived the project, performed the geochemical analysis, and interpreted the data. L.F.O. performed the TGA-MS experiments, C.L.B. performed the Raman analysis and supported the SEM analysis, and L.R. performed sample preparation. In addition, all authors contributed to the writing and editing of the manuscript text and figures.

## Competing interests

The authors declare no competing interests.
