## [Peer Review File · Nature Communications]

REVIEWER COMMENTS

Reviewer #1 (Remarks to the Author):

I appreciate the opportunity to read and review this manuscript. The experiment and results are novel and noteworthy, and deserve to be published following revisions of some major and minor points as I outline below. Most of these comments are related to improving the context of this work in relation to Phaethon and the other recent works that address the release of volatiles via extreme heating. The experimental methodology in this work is sound and well-presented in the supplementary sections.

Major Comments

- Recently, Zhang et al. 2022 (doi:10.3847/PSJ/acc866) suggested that sodium emission is solely responsible for Phaethon's comet-like tail and perihelion activity. This work should be mentioned somewhere in the introduction and in the context of the results of this work. I also strongly encourage the authors to report the detections of Na (or an upper limit) in their results table, as this information is pertinent to the wider discussion regarding Phaethon's activity.

- Phaethon has been shown to be spectrally similar to CY meteorites (more than heated CMs), as shown by MacLennan & Granvik (2024). This can be mentioned somewhere in the introduction (e.g. line 46) and throughout the work (lines 241 & paragraph at 260), as many of the discussion items in that paper closely parallel that of this work (i.e., the gas production estimates from decomposition, and timing of the activity). Please also see the next point about sulfides. M&G also found spectral evidence of calcium and magnesium hydroxide, which they explain as being a (back) reaction product of decomposed carbonates with water vapor. Is there any evidence (or none) from the heating experiments that this scenario is feasible?

- A high sulfide abundance was identified in Phaethon's mid-IR spectrum (MacLennan & Granvik 2024). What does this imply (if anything) about the heating that happens at or beneath the surface? This work showed sulfur being depleted in the samples, but remained concentrated in their rims (where remote sensing is most sensitive to). Can the authors speculate as to whether the high sulfide concentration from repeated heating cycles can properly explain the IR observations?

- The authors mention how their study is unique to other heating experiments in that their heating experiments are performed cyclically. The details are given in the supplementary section, yet the discussion in the main article is lacking any reflection on this aspect of the study. For example, there can be more added to the discussion about the differences shown by the various heating experiments as shown in Table 2. Are the findings (e.g. textural differences or elemental distributions) from the cyclic heating experiments much different than those found from the single-heating experiments?

- I couldn't find any mention of the origins of the water vapour emission, which is presumably due to phyllosilicate dehydration and/or decomposition. This was the most abundant volatile that was detected in the experiments. Can the authors say anything to the role of phyllosilicate decomposition into olivine? Was there any textural evolution of phyllosilicates seen in the SEM images?

Abstract

Revision of the abstract may be necessary following the revisions.

Introduction

The first sentence seems out of place in relation to the rest of the paragraph and the paper as a whole. I suggest removing it and starting the paper more focused on the heating of small bodies.

Line 35: Consider using Jones et al. (2018; doi:10.1007/s11214-017-0446-5) as a reference for heating of comets that approach the Sun.

Line 43: Zhang argues, convincingly, that the tail is not "dust" but made of sodium atoms.

Line 46: It should also be mentioned that MacLennan & Granvik (2024) identified a meteorite analog and several mineral species of Phaethon using mid-infrared spectral data. This spectrum DOES show diagnostic features, as opposed to the reflectance spectra already mentioned. The thermal decomposition process was an interpretation that came from their result.

Line 56: Is "ground truth" used as a verb here? Would "test ground-truth models" is more clear to the reader.

Line 60: "cycles which reflect".

Line 64 & 70: emission behaviour -> activity. As "emission" can be confused with infrared emission, or could refer to dust emission equally as volatile emission.

Line 67: Should Murchison be mentioned here already?

Line 69: please define "SEM"

Results

Line 75: Variability in the amount of water that was lost?

Paragraph at line 80: What is meant by "double heating", "stepped heating", and does "heating ramp" refer to one of these and not the other? These terms can be briefly defined when they are first mentioned to save the reader the trouble of searching though the methods/supplementary information.

Discussion

Equations 1 & 2: Perhaps it would be beneficial to include the reactions that lead to the production of water vapor and carbon dioxide somewhere, or provide a reference.

Line 145: Is there a ballpark estimate for the increase in permeability? Is it by a few factors or orders of magnitude, for example?

Line 150-151: Are the depleted rims large when compared to the chip size?

Line 238: "hydrated C-type asteroids" can be changed to "primitive asteroids" to refer to all asteroid that are likely carbonaceous chondrite composition (Phaethon is B-type asteroid).

Paragraph at 260: It should be mentioned here that Phaethon's spectrum matches with the CY chondrites (MacLennan & Granvik 2024). Even though M&G effectively ruled-out heated CMs in favor of CYs, the composition of the CM samples used here are quite suitable for Phaethon (even though differences in sulfide abundance or Ca/Mg may exist).

Supplementary Information

Supp. Figure 4. What are the green and red curves? Adding labels in the figure next to the curves would make it easier for the reader to understand the figure.

Supp. Figure 5. A reference image of this sample before heating would better show evidence of fractures.

Supp. Figure 9. It appears that the caption has reversed the positions of the TGA and mass spectrometer.

Supp Table S2. The caption says the rows are ordered by S abundance, so should the Unheated and Sample B rows be swapped?

Reviewer #2 (Remarks to the Author):

I attached a PDF with the review.

Response to reviewers document: Rapid heating rates define the volatile emission and regolith composition of (3200) Phaethon.

Suttle, M.D.^{1,2}, (*corresponding author*), Olbrich, L. F.³, Bays, C.L.^{2,4}, Riches, L.¹.
martin.suttle@open.ac.uk, lorenz.olbrich@materials.ox.ac.uk, charlotte.bays@nhm.ac.uk,
liza.riches@open.ac.uk.

¹School of Physical Sciences, The Open University, Walton Hall, Milton Keynes, MK7 6AA, UK.

²Planetary Materials Group, Natural History Museum, Cromwell Road, London, SW7 5BD, UK.

³Department of Materials, University of Oxford, Parks Road, Oxford OX1 3PH, UK.

⁴Department of Earth Sciences, Royal Holloway University of London, Surrey TW20 0EX, UK.

NOTE: Reviewer comments are shown in black text, author responses are shown in blue. *Italic text* is used to represent quotes from the manuscript and underlined text represents new additions, added to the revised version.

Reviewer #1: I appreciate the opportunity to read and review this manuscript. The experiment and results are novel and noteworthy and deserve to be published following revisions of some major and minor points as I outline below. Most of these comments are related to improving the context of this work in relation to Phaethon and the other recent works that address the release of volatiles via extreme heating. The experimental methodology in this work is sound and well-presented in the supplementary sections.

Major Comments:

- Recently, Zhang et al. 2022 (doi:10.3847/PSJ/acc866) suggested that sodium emission is solely responsible for Phaethon's comet-like tail and perihelion activity. This work should be mentioned somewhere in the introduction and in the context of the results of this work. I also strongly encourage the authors to report the detections of Na (or an upper limit) in their results table, as this information is pertinent to the wider discussion regarding Phaethon's activity. – Good point! We've cited the Zhang et al. 2022 study (and a similar Abe et al. 2020 study on Na in Geminid meteors). This was added in the introduction where we mention Na-emission as potential volatile gas source. Most significantly, we've added data to the results on Na abundances/distribution, a new Na map supplementary figure and a new subsection of the discussion covering the role of Na as a volatile source in the manuscript.

In the discussion section we have quantified the detection limit for Na in our experiments. Note, we are confident that there was negligible Na release from all the samples, we would have observed a pattern of a reoccurring peak at \$mz=23\$, which was not the case.

Although it is not possible to provide a single lower detection limit value in wt.% for all experiments since this value was dependent on the total mass measured loss over the experiment. The total ion current* \$mz\$ was set equal to the observed mass loss in the TGA. Thus, for section 1 of sample A \$3 \mu\text{As}\$ (around \$70 \mu\text{As}\cdot\text{u}\$ ) equalled \$\sim 5.7\$ wt.%, while in section 2 of sample A around \$1 \mu\text{As}\$ (around \$29 \mu\text{As}\cdot\text{u}\$ ) equalled to \$\sim 5.0\$ wt.%. Our detection limit was \$0.0005\$ nA (outlined in the "online methods, heating experiments" section). For sample A, section 1 with a window of \$94\text{min} = 5640\$ s, the maximum amount of Na release would be \$2.82\$ nAs. Multiplied by the atomic mass of Na (\$23\text{u}\$ ), this corresponds to an integral of \$0.013 \mu\text{As}\cdot\text{u}\$ which is \$0.018\%\$ of the total ion current integral and, thus, corresponds to a mass loss of \$0.001\$ wt% over this window.

Repeating this calculation for section 2 of sample A yields: $0.005 \text{ nA} \cdot 8640 \text{ s} = 4.32 \text{ nAs}$, $4.32 \text{ nAs} \cdot 23 \text{ u} = 0.1 \text{ } \mu\text{As} \cdot \text{u}$, $0.1 \text{ } \mu\text{As} \cdot \text{u} / 29 \text{ } \mu\text{As} \cdot \text{u} = 0.34\%$ of total ion current integral, 0.34% of $5 \text{ wt}\% = 0.017 \text{ wt}\%$.

Taking this approach, the detection limit for Na in our experiments lies at approximately **0.02wt%**.

- Phaethon has been shown to be spectrally similar to CY meteorites (more than heated CMs), as shown by MacLennan & Granvik (2024). This can be mentioned somewhere in the introduction (e.g. line 46) and throughout the work (lines 241 & paragraph at 260), as many of the discussion items in that paper closely parallel that of this work (i.e., the gas production estimates from decomposition, and timing of the activity). – That’s correct, this work builds on the discoveries of MacLennan & Granvik 2024. This study is now more widely referenced and we directly explain how our work extends beyond the M&G study through new additions to the discussion (see discussion subsections: An experimentally constrained model for the regolith composition and activity of Phaethon AND Is Phaethon a CY or CM chondrite?

M&G also found spectral evidence of calcium and magnesium hydroxide, which they explain as being a (back) reaction product of decomposed carbonates with water vapor. Is there any evidence (or none) from the heating experiments that this scenario is feasible? – Yes, in the original submission we discussed portlandite formation and why it did not occur in our experimental samples. This reaction route first suggested in the context of meteoritic materials by Haberle and Garvie (2017). We’ve now added reference to the M&G as well and the likelihood of portlandite on Phaethon.

- A high sulfide abundance was identified in Phaethon's mid-IR spectrum (MacLennan & Granvik 2024). What does this imply (if anything) about the heating that happens at or beneath the surface? This work showed sulfur being depleted in the samples, but remained concentrated in their rims (where remote sensing is most sensitive to). Can the authors speculate as to whether the high sulfide concentration from repeated heating cycles can properly explain the IR observations? – Covered in discussion subsection: “Is Phaethon a CY or CM chondrite? In short, on Phaethon the high Fe-sulphide abundance occurs due to passive enrichment, while in the CYs the high Fe-sulphide abundance is either accretionary or a result of S-rich fluid during aqueous alteration and so not formed in the same way. Therefore, we argue the similar mineralogy do not necessarily argue for a direct match between these groups.

- The authors mention how their study is unique to other heating experiments in that their heating experiments are performed cyclically. The details are given in the supplementary section, yet the discussion in the main article is lacking any reflection on this aspect of the study. For example, there can be more added to the discussion about the differences shown by the various heating experiments as shown in Table 2. Are the findings (e.g. textural differences or elemental distributions) from the cyclic heating experiments much different than those found from the single-heating experiments? – This is a good point and textural/petrographic comparison of the cyclic heating samples against single heating and natural metamorphosed CMs is of interest to the meteoritics community. However, this was covered well in Patzek et al. 2024 and would be beyond the scope of this highly focused work. The AE stressed that the R1 revision should be concentrated on the insights into Phaethon’s volatile emission as opposed to the wider implications to meteoritics or other asteroids. We have therefore not expanded upon these points in the R1 revisions.

- I couldn't find any mention of the origins of the water vapour emission, which is presumably due to phyllosilicate dehydration and/or decomposition. This was the most abundant volatile

that was detected in the experiments. (From the initial submission, discussion: "...and released gases from phyllosilicates (H₂O)" (line 188) AND "...water released from decomposing phyllosilicates" (line 200) AND "...due to the rapid liberation of H₂O from phyllosilicates" (line 209)).

Can the authors say anything to the role of phyllosilicate decomposition into olivine? Was there any textural evolution of phyllosilicates seen in the SEM images? – No, we did not see any unusual texture features in the matrix arising from dehydration. The thermal decomposition of phyllosilicate and recrystallization as secondary olivine is a well-established solid-state process that faithfully preserves the texture of the fine-grained matrix. E.g., Suttle et al. 2017 The thermal decomposition of fine-grained micrometeorites, observations from mid-IR spectroscopy. *Geochimica et Cosmochimica Acta*, 206, pp.112-136.

Introduction.

The first sentence seems out of place in relation to the rest of the paragraph and the paper as a whole. I suggest removing it and starting the paper more focused on the heating of small bodies. – Changed.

Line 35: Consider using Jones et al. (2018; doi:10.1007/s11214-017-0446-5) as a reference for heating of comets that approach the Sun. – Thanks for the reference, a useful resource. However, on line 35 I am referring explicitly to near-Sun asteroids and not comets, so this reference is not appropriate here.

Line 43: Zhang argues, convincingly, that the tail is not "dust" but made of sodium atoms. – noted, I've removed the term dust from this sentence, reflecting our current ambiguity of the composition of the tail.

Line 46: It should also be mentioned that MacLennan & Granvik (2024) identified a meteorite analog and several mineral species of Phaethon using mid-infrared spectral data. This spectrum DOES show diagnostic features, as opposed to the reflectance spectra already mentioned. The thermal decomposition process was an interpretation that came from their result. – Addressed in the subsections: An experimentally constrained model for the regolith composition and activity of Phaethon AND Phaethon – a spectral link to the CYs or the CMs?

Line 56: Is "ground truth" used as a verb here? Would "test ground-truth models" is more clear to the reader. – changed.

Line 60: "cycles which reflect". – changed.

Line 64 & 70: emission behaviour -> activity. As "emission" can be confused with infrared emission, or could refer to dust emission equally as volatile emission. – Changed.

Line 67: Should Murchison be mentioned here already? – revised. Here we now state: "Building on previous studies, we set out to investigate the activity of Phaethon through an experimental heating study. We performed repeated heating-cooling cycles using a Phaethon analogue material (Table. 1)" and the next paragraph explains why Murchison was chosen as the most appropriate Phaethon analogue.

Line 69: please define "SEM" – defined.

Line 75: Variability in the amount of water that was lost? – changed.

Paragraph at line 80: What is meant by "double heating", "stepped heating", and does "heating ramp" refer to one of these and not the other? These terms can be briefly defined when they

are first mentioned to save the reader the trouble of searching through the methods/supplementary information. – Agreed! Clarified.

Equations 1 & 2: Perhaps it would be beneficial to include the reactions that lead to the production of water vapor and carbon dioxide somewhere, or provide a reference. – Added equation for carbonate decomposition as relevant for the back-reactions to Ca-sulphates/hydroxides. We've not added the phyllosilicate decomposition reaction since the specifics of the resulting solid phases (olivine) are not crucial to the conceptual model covered here.

Line 145: Is there a ballpark estimate for the increase in permeability? Is it by a few factors or orders of magnitude, for example? – Unfortunately, we could not find any permeability measurements or models for heated/metamorphosed carbonaceous chondrites in the literature. Qualitatively we can see that permeability increases but I cannot quantify this at the present stage. BUT permeability must limit gas escape over the length-scales and durations of our experiments this is because released volatile gases do not escape during the first heating event, as outlined in the discussion and demonstrated by the emission of gases during second heating at the same temperature as before and by the observed volatile element migration.

Line 150-151: Are the depleted rims large when compared to the chip size? – From the results section: "...and a thin (30-60 μm) sulphur depleted zone along the chip's perimeter was observed (Fig. 2B). Chips affected by >1 heating cycle had thicker sulphur depleted rims (typically >200 μm).". Exposed sections are mm-scale, with long-axis dimensions typically between 2.5 mm. We've added into the results section the following clarification "(exposed surface areas of approx. approx. 10 mm²)".

Line 238: "hydrated C-type asteroids" can be changed to "primitive asteroids" to refer to all asteroid that are likely carbonaceous chondrite composition (Phaethon is B-type asteroid). – changed.

Paragraph at 260: It should be mentioned here that Phaethon's spectrum matches with the CY chondrites (MacLennan & Granvik 2024). Even though M&G effectively ruled-out heated CMs in favor of CYs, the composition of the CM samples used here are quite suitable for Phaethon (even though differences in sulfide abundance or Ca/Mg may exist). – Addressed in dedicated discussion subsection.

Supp. Figure 4. What are the green and red curves? Adding labels in the figure next to the curves would make it easier for the reader to understand the figure. – revised.

Supp. Figure 5. A reference image of this sample before heating would better show evidence of fractures. – revised.

Supp. Figure 9. It appears that the caption has reversed the positions of the TGA and mass spectrometer. – Yes, well spotted, changed, thanks.

Supp Table S2. The caption says the rows are ordered by S abundance, so should the Unheated and Sample B rows be swapped? – Corrected.

Reviewer #2 (Markus Patzek): This paper reports on an experimental study to understand the volatile emissions of Near-Sun. asteroid Phaethon during perihelion by cycling CM chondrite fragments to different temperatures and rates and measure their degassing characteristics (amount and species). They propose a model to explain the observed comet-like activity, by

the rapid heating and accompanied cracking providing new pathways of volatile degassing in the rocks.

Impressions: The manuscript is well written and discusses the general topic, motivation and analytical approach used. It describes the proposed reaction pathways during heating and cooling phases and provides the reader with a model able to describe the observations made on the experimentally heated and cycled CM chondrite lithologies. I have some minor (major?) comments on some aspects of the paper that are presented down below. Overall, I recommend the publication of this work. I would like to see the replies to the review (because I asked some specific questions) afterwards.

I enjoyed reading your manuscript and I think it is in a very good shape and I am happy to see more experiments focusing on cycling scenarios of planetary materials. I have some general points I would like to see discussed (at least in supplementary material and to a certain degree also in the main text):

- What drives the selection of cycle number? I understand, that there are some analytical problems of capacity and workload; However, why couldn't you measure the first 5 cycles, then cycle 15x and measure again 1-3) to understand how much the cycling itself changes the volatile emission. From our work we saw not much happening on the surface at lower T_{max}, but this does not necessarily mean, that nothing happens at your T_{max}. – The selection of experimental parameters aimed to meet a compromise between modelling the heating regime of Phaethon (as closely as possible) and pragmatic constraints. We were limited by time available on the instrument and by duration of experimental runs (technician time). I agree that it would have been interesting to run longer experiments (more cycles and TGA+MS for some of those later cycles) but this was not possible. However, analysis of the experimental rock chips under SEM and Raman after the 8x cycle provided crucial data on how the texture, mineralogy and volatiles have evolved and migrated over many repeated heating events.
- Follow-up on the first one: Impact of continuous cycling. Yes, the first things happen very early, but if nothing further happens, the surface might be passivized by a dehydrated and thermally decomposed layer of regolith on the asteroid that insulates the “interior” portions. – Ultimately yes this is the end state of any material subject to repeated cyclic solar radiant heating. All volatile gases that can be emitted are released within the volume affected by heating and the surface is inert until it can be renewed by shedding (or a similar mechanism). Our experiments show that there is much activity early on (1st and 2nd cycle) while the 8x heated sample B provides a perspective on more mature repeated heating. We would expect a sample heated to say 80x to be inert. We've added in some clarification to the end of the discussion to highlight this point:
“Ultimately the Fe-sulphides will fully decompose to form Fe-metal at which point comet-like activity will cease until fresh unheated chondritic surface is exposed by dust shedding, tidal fracture or another resurfacing mechanism.”
- Maybe you should leave a word on the size of the particles, since this certainly plays a role on the formation of the stress fields (and therefore how the cracks develop) in the interior and how you thermally shock the material (aspect of thermal skin depth). – We've added in details on the mass, size and amount of exposed surface area studied for the chips in this work: “Whole rock chips (36-60 mg and ~3-4 mm diameter)” AND “(exposed surface areas of ~10 mm²)”.
- How does the short rotation rate of Phaethon play a role? The temperature in perihelion is high, but maybe the cycling (DT) is happening on much smaller amplitude, because the baseline temperature is generally very high due to reaching a higher baseline temperature? Would love to see more discussions in this respect. – We've stated the

rotational period of Phaethon in the introduction.

The study of MacLennan et al. 2021 explored the thermal history of Phaethon. This work includes plots of Phaethon's surface temperature and rates of temperature change as a function of latitude (their Fig.8). These plots were used in the selection of heating regimes defined in our study. They show heating rates of 10-25 K/min and peak temperatures of ~600-1000 K (~330-730 °C).

In the revised manuscript the introduction states: *"Phaethon has a rotational period of 3.604 hours (Hanuš et al. 2016), peak temperatures during perihelion reach up to ~730 °C (~1000 K) and heating/cooling rates are on the order of 10-25 °C/min (MacLennan et al. 2021). Our selection of peak temperatures 500-750 °C (773-1023 K) and heating/cooling rates between 2-20 °C/min was designed to directly mirror the range of heating scenarios applicable to surface and shallow subsurface environments on Phaethon"*.

- Also, what does the time before perihelion do the surface material, i.e., what effect does this have on the degassing profile? T_{max} will gradually increase with Phaethon (or any other NSA, "near Sun asteroid") getting closer to the Sun. Of course there will be no significant dehydration below the dehydration temperature of the phyllosilicates, but maybe the cycling is still doing something to the material. Considering geologic timescales, the T_{max} gradually increases with each orbit around the sun, a potential passivation (degassed material) took already place. I think, that the renewal of surface material is constantly required (as known) and the main follow-up question is, whether the gas drag of the degassing after gradually building-up temperatures is able to lift particles over the escape velocity. – These are interesting questions(!), particularly the points about surface renewal and the lifetime of this process (which is necessary to expose fresh, unheated material). I have added some discussion covering the role of resurfacing:

"Near-identical magnitude brightening, reoccurring each orbit has been observed around perihelion (Hui and Li, 2016). Recent high-resolution thermal IR imaging constrained an upper limit for the quantity of dust released from Phaethon (<14 kgs⁻¹). This is estimated as 50x less than the rate required to sustain the Geminid meteor stream (Jewitt et al. 2019). Furthermore, Zhang et al. (2023) argued that the comet-like tail observed extending from Phaethon (Jewitt and Agarwal, 2013) is unlikely to be a dust tail but, instead a (Na) ion tail. Thus, the role of surface renewal by dust shedding in the current era seems minimal yet Phaethon continues to demonstrate comet-like activity (gas emission). Continued activity from a surface that has been repeatedly heated over thousands of years (>20,000 years [MacLennan et al. 2021]) is what makes explaining the activity of Phaethon challenging...."

However, points on gas drag removing material are not able to be answered based on the data gained in these experiments so we have limited our discussion to focus on the gas emission behaviour.

- In general, I found the different samples and applied heating profiles ("shape" and T_{max}) a bit confusing during reading. I understand the different motivation and have no direct suggestion. – We've added in some additional clarification regarding what is meant by the double heating and stepped heating profiles (in both the results and Table 1 caption).
- Is there a difference between the depth of the S-depleted zones in the samples cycled under vacuum? Asking because of the proposed back-reaction of the sulfur-bearing gases on the retrograde cooling, which are potentially more efficiently driven out under vacuum? – I've added additional details on S-rim depletion and thickness into the results section and added in a small section of supplementary discussion covering these points.

I hope you can follow my comments. In case something is unclear or needs further discussions, I am happy to help also via mail.

REVIEWERS' COMMENTS

Reviewer #1 (Remarks to the Author):

Thanks for addressing all of the comments from the first review round. I enjoyed the expanded discussion points which better place the results in the wider context of Phaethon and the related works ahead of the DESTINY+ mission.

Reviewer #2 (Remarks to the Author):

Thanks for the answers. I am happy with the changes and looking forward to publication.